# Factors Affecting Organisations' Adoption Behaviour toward Blockchain-Based Distributed Identity Management: The Sustainability of Self-Sovereign Identity in Organisations

Sarah Mulombo Mulaji *[ID] and Sumarie Roodt

Department of Information Systems, University of Cape Town, Rondebosch, Cape Town 7701, South Africa
* Correspondence: mljsar001@myuct.ac.za

**Abstract:** Blockchain-based Distributed Identity Management (BDIDM) can enhance sustainable identification and authentication of users on organisations' digital systems. But there is not a clear consensus on how organisations perceive the value proposition of such technology, nor what might affect their adoption behaviour toward it. This research explains how technological, organisational and environmental (TOE) factors affect organisations' adoption behaviour toward BDIDM. This study aims to determine the most critical factors affecting the behaviour while assessing the effectiveness and appropriateness of the model involved, i.e., TOE-BDIDM. Online questionnaires are used to survey 111 information and cybersecurity practitioners within South African organisations. The analysis combines binary logistic regression modelling, Structural Equation Modelling of the measurement model, and chi-squared tests. The results suggest TOE factors positively or negatively affect adoption behaviour. The behaviour is significantly affected by technology characteristics, i.e., BDIDM's disruptive nature, and is associated with Blockchain type. Indeed, the majority of participants intended to recommend BDIDM to their organisations yet paradoxically preferred private-permissioned blockchain the most, revealing resistance to decentralised and privacy-preserving BDIDM models like Self-Sovereign Identity (SSI). The latter might be utopian or unsustainable for organisations. TOE-BDIDM was found relatively appropriate and effective but arguably 'incomplete' for explaining the adoption of disruptive technologies like SSI in organisations. TOE should extend to TOEU by including the User factors.

**Keywords:** blockchain for enterprise; distributed identity management; centralised and decentralised IDM; predicting organisations' adoption behaviour; permissioned and permissionless blockchains; resistance to self-sovereign identity; SSI sustainability; sustainable identification and authentication; disruptive technology characteristics; Technology-Organisation-Environment model; TOE appropriateness and effectiveness; Information and Cybersecurity Practitioners Survey

## 1. Introduction

Identity management (IDM) is the first security barrier of a digital system that consists of *identification* and *authentication* of users to mitigate security breaches. Identification labels each user with an identifier, while authentication allows them to prove they are who they claim to be [1]. However, just about all authentication methods (such as passwords, biometrics, tokens, etc.) can be compromised since they have known vulnerabilities that could be exploited [1]. When users' credentials are compromised, the security of every system relying on them to authorise access is also compromised [2]. Multi-factor authentication is perceived to overcome this issue and offer better identity proofing [3]. Nevertheless, organisations still experience data breaches. Personal identifiable information (PII), from "billions of email account details to millions of credit and identity data", can now be found on the internet [4]. A growing tendency suggests that the centralised architecture used in today's IDM systems might be problematic [5].

Traditional centralised IDM systems embed the critical vulnerability of a single point of failure (SPOF) [6] because they use a server to store users' credentials and related identity data. This data is exposed when the server is compromised, violating the security of the system on the one hand and users' privacy on the other hand [7]. Balancing security and privacy in organisations is an increasingly challenging task, if not an IDM dilemma. Organisations must know "as much as possible about their (potential) customers" as part of their customer diligence requirements [8] (p. 22). Yet, they need to preserve users' privacy in compliance with government regulations, such as the Protection of Personal Information Act (POPIA) in South Africa.

Meanwhile, internet growth has resulted in dozens of digital accounts per user for different online services they subscribe to, forcing many to adopt insecure behaviour like reusing the same credentials with various services [9]. Others have been using weak credentials, so they are easier to remember, making it easier for imposters to guess them [2]. Secure and reliable IDM has also proven to be among "the greatest challenges facing cloud computing today" [10] (p. 724), being essential to achieving the requisite 'secure cloud' [11] (p. 91). 'Secure IoT' (Internet of Things) is another crucial IDM challenge, requiring organisations to effectively identify and authenticate interconnected 'things' interacting with people on a digital system [12,13]. 'Things' include software, smartphones, robots, automobiles, appliances, entertainment devices, etc.

Innovative IDM systems like BDIDM are emerging to deal with some of the above IDM challenges [6]. Despite its immaturity [14], BDIDM is reported to have no SPOF vulnerability [15] thanks to its decentralised and disintermediated architecture. Typical BDIDM models like SSI are claimed to be privacy-preserving [7] and dissolve the need for multiple accounts by enabling identity interoperability among different online services [16]. SSI might also facilitate 'secure cloud' and 'secure IoT' [4] in the form of distributed ID-as-a-Service [17]. Thus, organisations might consider adopting BDIDM to avoid potential competitive disadvantages.

Beyond organisations' legitimate need to address IDM issues, there is more that might affect the likelihood of them adopting such innovation. The technology, organisation, and external environment (TOE) model depicts a comprehensive context of enterprise adoption of innovation. TOE is claimed to explain how an organisation "identifies the need, searches, and adopts new technologies" [18] (p. 232). Therefore, this research investigates how TOE factors might predict organisations' adoption behaviours toward BDIDM to determine the most critical predictors and examine their effect. TOE theory is also claimed to have "broad applicability" across various technological, industrial, and national/cultural contexts [18] (p. 151). This study also seeks to verify this claim by assessing TOE-BDIDM's effectiveness and appropriateness in investigating the phenomenon of organisation's adoption behavior in the context of IDM.

The few studies on blockchain adoption mainly focus on its individual rather than enterprise adoption. This study contributes to the body of knowledge about perceptions of blockchain applications in the enterprise context to address the lack of empirical research on the topics. Knowing how organisations perceive the value proposition of BDIDM will inform scientific enquiries for more realistic blockchain design theories, which will in turn impact the practice. The findings could also inform IDM policymaking in organisations and inspire future research. Theoretically, this study contributes to extending the TOE theory. It advocates for a more comprehensive theory, TOEU, that includes the User context to better explain organisations' adoption behaviour toward disruptive technologies like SSI. This enterprise-level theory of innovation adoption has not been given much attention like other related theories yet.

The rest of the introduction gives additional background necessary to understand the topic and the TOE-BDIDM model.

### 1.1. The 'Ideal' BDIDM Model for Organisations

The implementation of any blockchain application includes answering three fundamental questions: (i) who can join the network (public vs. private); (ii) whether a validator will be needed (permissioned vs. permissionless); and (iii) what type of consensus protocol (such as Raft, Federated, Proof of Authority, Proof of Work, Proof of Stake, etc.) will regulate interactions among participants [19]. Combined answers to these questions result in three types of blockchains: public permissionless, public permissioned, and private permissioned [20]. In BDIDM terms, public permissionless blockchains are not restricted. Anyone can join to get their DID and self-manage it. They have full control over it and can use it wherever they wish. Public permissioned blockchains are somewhat restricted. Some users can join to get their DIDs and self-manage them as they also have full control over them. Private permissioned blockchains are completely restricted to known and trusted users only. The validator known as "Trust Anchor" [21] (p. 46) has control over users' DIDs and could, for instance, block a particular DID if necessary.

Some researchers support the notion that public permissioned blockchains are ideal for implementing typical BDIDM models like SSI [16] since they are perceived as more balanced versions of blockchains [22]. They are more decentralised, scalable, efficient [23], and offer "privacy protection and high transparency" [24] (p. 21573). Others argue that private permissioned blockchain might be the perfect implementation for 'enterprise BDIDM' because it endorses a service-centric approach by giving total control of the system to the identity provider, i.e., Trust Anchor [21]. However, a private permissioned blockchain would not differ much from traditional centralised models, from which organisations might want to move.

A different view suggests that the choice of the ideal BDIDM implementation ought to depend on trust assumptions [25]. From the outsider-threat perspective of cybersecurity theory [1], this implicit trust means that any type of BDIDM would be unnecessary for 'trusted users' (staff members accessing corporate systems from the intranet); permissioned BDIDM would make sense for 'semi-trusted users' (clients, suppliers, partners, etc., accessing corporate systems from the extranet); and permissionless BDIDM would be perfect for 'untrusted users' (visitors or any unknown user accessing corporate systems from the internet) [25]. However, the National Institute of Standards and Technology (NIST) highlights the new tendency to shift from this traditional implicit trust framework to a Zero Trust (ZT) security architecture. ZT tends to support the insider-threat cybersecurity theory that indirectly advocates for even more radical BDIDM, like SSI. ZT theory suggests that all entities ought to be untrusted, and prevented from accessing corporate systems by default: all should rigorously prove the authenticity of their identity in order to gain access [26].

These diverging views around the ideal blockchain implementation for enterprise BDIDM amplify the challenges of adoption and suggest the criticality of blockchain type on BDIDM adoption decision making in organisations. The findings shall reveal which blockchain type organisations might prefer the most and whether that is associated with their adoption behaviour toward BDIDM.

### 1.2. The Sustainability of the SSI Model in Organisations

SSI is a BDIDM model famous for maintaining users' privacy and thus enhancing sustainable management of identities on digital systems. Organisations' use of a privacy-respectful mechanism in identifying and authenticating users could mitigate data breaches, especially PII leaks [27,28]. However, there are controversies about SSI sustainability for organisations since the traditional centralised logic of IDM might be hostile to privacy. SSI adoption might involve solving the conflict between privacy and trust [29].

SSI for enterprise could be illustrated with the scenario of a newly recruited employee. The new employee, Alice, would not need to disclose any PII to their employer, nor create any digital account with them. Thus, there are no additional password/s for Allice to recall. Alice would simply invite their new employer to verify their already existing DID to access the corporate digital assets. Consequently, the employer has no control over that

DID and Alice's privacy is preserved [30]. From the perspective of traditional information security principles, this scenario is disruptive for organisations. An organisation would not trust Alice's DID, because it is external [25]. The organisation would question whether participants in the blockchain storing Alice's DID are trustworthy. The organisation would be concerned about potential risks involved when Alice's DID gets hacked, or wonder whether an imposter was behind Alice's DID to spy on the company's business [1]. It would be nearly unacceptable for the organisation to lose control over Alice's DID since it is used to access its confidential information [2].

SSI opponents stress blockchain's weaknesses at its endpoints [31]. The PII anonymisation not only means that there is no central authority to block an account in case of identity theft or misbehaviour but also that "each user must themselves safeguard against forgetting (or losing) the private key" [32] (p. 5). SSI is risky because the user is the only one responsible for managing all the cryptographic keys protecting their DIDs [7]. Some researchers even question whether further adoption of blockchain-based solutions ought to be encouraged and whether the overall potential for change "could be [a] net positive" [33] (p. 447). However, others recommend proactive planning instead of fear of the unknown, arguing that "reluctance to adopt disruptive technologies may be a significant competitive disadvantage for an organisation" [14] (p. 34).

Indeed, blockchain technology offers a paradigm shift in developing next-generation cybersecurity barriers. Blockchain-based solutions like BDIDM ensure data integrity due to cryptography and non-reliance on passwords. A blockchain is, by design, resilient to SPOF and "assumes the presence of adversaries in the network", making tampering with it extremely difficult (but not impossible) [6] (p. XIII). The identity self-management feature of SSI could lead to reduced costs of transactions for both users and organisations [7]. Users benefit from the reduced costs of identity theft and PII leaking. Organisations benefit from the exemption from protecting any PII and replicating it among interested services, reducing related costs of privacy infringements. The cost savings in password management ranges in the millions of dollars [34]. Yet, one would argue that organisations would still prefer to pay these costs than lose control over users.

These divergences make BDIDM adoption in organisations even more daunting. The study findings shall reveal the extent to which organisations are likely to adopt BDIDM. The findings shall especially reveal organisations' adoption behaviour toward SSI. As mentioned earlier, other factors are involved in predicting organisations' adoption behaviour toward BDIDM. This study chose to classify them all according to the TOE theory as technological, organisational, and environmental factors. These factors form the basis of the TOE-BDIDM model.

### 1.3. The TOE-BDIDM Research Model

The TOE-BDIDM model in Figure 1 is a particularised version of the TOE model initially developed by Tornatzky and Fleischer in 1990 as part of *The Processes of Technological Innovation* then updated later by Jeff Baker in 2011 [18]. The model is operationalised to guide empirical investigation of the enterprise BDIDM adoption phenomenon [30]. The present study (i) develops ten alternative hypotheses, $H_{a1}$ to $H_{a10}$, represented by the arrows between the independent variables (The nine first-order constructs) and the dependent variable (adopt or not adopt BDIDM) as shown in Figure 1. The study then (ii) tests the associated null hypotheses $H_{01}$ to $H_{010}$, and then (iii) assesses the TOE performance in the BDIDM context. The sub-sections of technological factors, organisational factors, and environmental factors detail the hypotheses' development to anticipate organisations' adoption behaviour toward BDIDM.

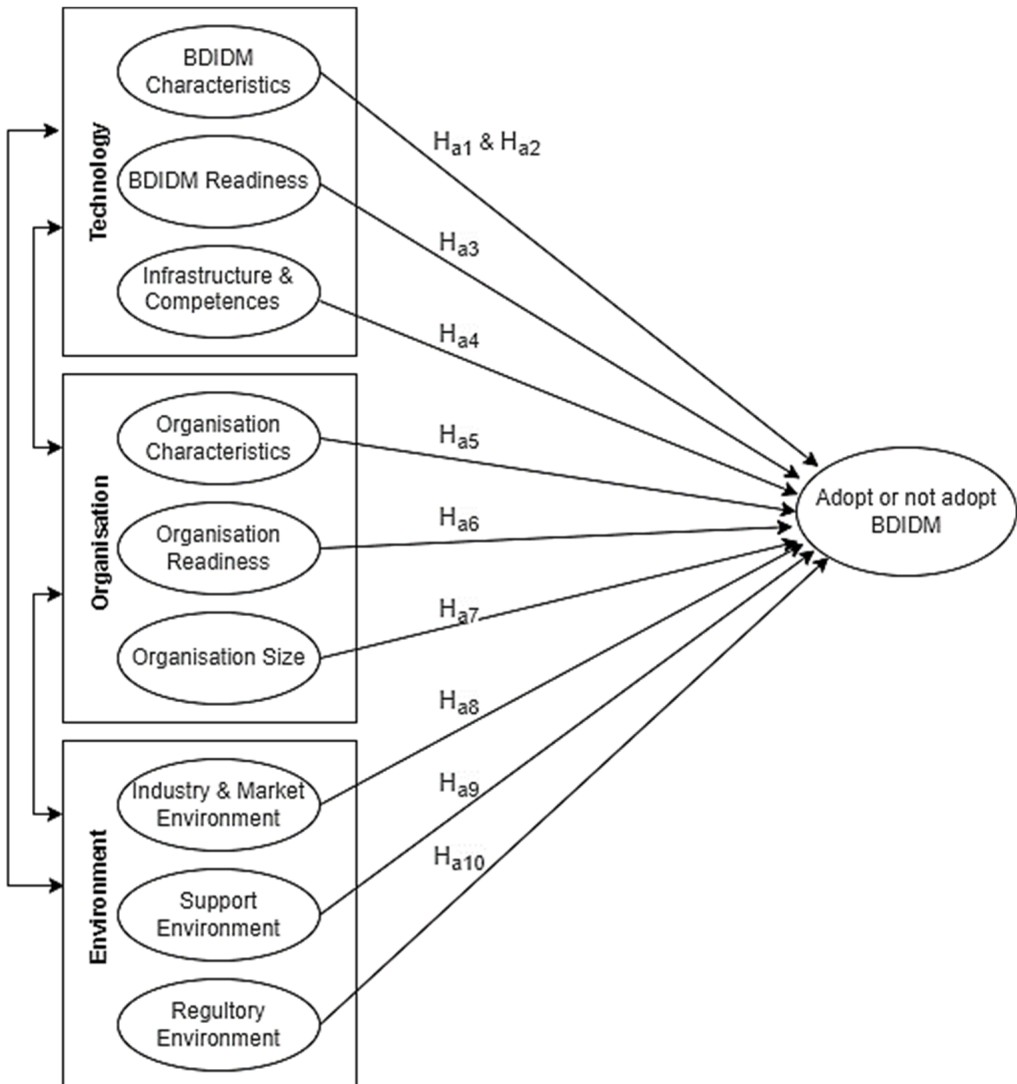

**Figure 1.** The TOE-BDIDM model.

1.3.1. Technological Factors

Technologies in use and those existing in the marketplace but not yet adopted can affect how organisations adopt an innovation [18]. In the case of BDIDM, legacy centralised technologies may constrain its adoption due to their incompatibility with a distributed architecture [21]. BDIDM might lead to a complete rebooting of the organisation's Information Technology (IT) infrastructure [35]. Since a poorly designed blockchain could be disastrous, its adequate implementation requires new competencies, which are still rare [36]. Thus, we posit that *IT infrastructure and competencies have a statistically significant effect on organisations' adoption behaviours toward BDIDM ($H_{a4}$).*

The market is now offering some BDIDM products, providing insight into BDIDM readiness in terms of its governance and standardisation [37]. However, there are inconsistencies in updating blockchain's fundamental rules [37]. Consensus-based rules involve all nodes, "which can be any device" [38] (p. 242). This seems problematic for organisations following centralised-IDM logic and could disrupt the accountability and management of the network [39]. Therefore, we hypothesise that *BDIDM readiness (for the enterprise context) has a statistically significant effect on organisations' adoption behaviours toward BDIDM ($H_{a3}$).*

Innovation's characteristics, i.e., the extent of the change it brings, also affect organisations' adoption decision making. BDIDM is a radical innovation. It does not increment or synthesise existing technologies but tends to reboot them in a radical manner

through distributed computing [40]. Such innovations produce "discontinuous change" and have higher adoption risks, yet can "enhance competitive standing in an organisation" [18] (p. 232). Technology characteristics relevant to BDIDM adoption include trialability, observability, compatibility, and complexity [41]. Another key BDIDM characteristic is the security construct. The CIA triad theory states that security balances confidentiality, integrity, and availability; thus the acronym CIA [1]. The Trust Service Framework (TSF) adds a fourth item of privacy [42]. This extension means that security without privacy is problematic, just as a four-legged table cannot balance if one leg is missing [30]. This principle seems to be what BDIDM, especially SSI, advocates for. All these dynamics lead to hypothesising that *BDIDM characteristics have a statistically significant effect on organisations' adoption behaviours toward BDIDM ($H_{a1}$).*

Blockchain types might be another essential facet of BDIDM characteristics. It seems there are underlying diverging views in the literature around the ideal BDIDM implementation for enterprises, which might signal significant differences among blockchain types associated with BDIDM adoption decision making in organisations. Therefore, we anticipate that *statistically, blockchain types are significantly different and associated with organisations' adoption behaviours toward BDIDM ($H_{a2}$).*

Another relevant technological construct is "technical know-how" [43] (p. 7), made up of the availability of skills, consultants, and vendors of BDIDM. However, this study chooses to view these items as external to the organisation and thus identifies them under environmental factors [18].

### 1.3.2. Organisational Factors

The organisational factors include a firm's characteristics that can affect organisations' adoption behaviour, typically through its structure and communication processes [44]. Organisational structures like formal and informal mechanisms linking different organisation units may promote innovation [18]. Organisations with an organic and decentralised structure may cope well with the BDIDM adoption phase. Those with formal reporting relationships, centralised decision-making, and clearly defined roles for employees may instead cope well with the implementation phase [18]. Communication processes or channels like "top management support" [45] (p. 1457) can either promote or constrain the adoption since it is key to preparing a corporate culture that welcomes change. The support would typically include describing the BDIDM value proposition, rewarding related initiatives, and building "a skilled executive team" that can cast a compellingly firm vision [46] (p. 233). Thus, we posit that *organisation characteristics have a statistically significant effect on organisations' adoption behaviours toward BDIDM ($H_{a5}$).*

Another organisational factor consists of organisations' readiness in terms of (financial) resources and awareness. BDIDM upfront implementation cost is perceived as high financially and requires highly skilled labour [36]. Blockchain technology was initially designed for "an investment rather than a traditional business use with an expected return on investment" [14] (p. 37). Moreover, organisations tend to be hostile to strongly privacy-flavoured BDIDM like SSI [45]. BDIDM's relative newness raises the importance of organisations' awareness, preparedness, and cultural adaptation [39] for such change as opposed to their "reluctance to change" or "fear of unknown technology" [14] (p. 37). We, then, hypothesise that *organisation readiness has a statistically significant positive effect on organisations' adoption behaviours toward BDIDM ($H_{a6}$).*

Organisation size is considered a minor organisational factor, as there have not been many empirical studies confirming its link to innovation adoption [18]. However, due to the implementation cost involved [14], SMEs may be less likely to adopt BDIDM than larger organisations [45]. Small businesses and startups would not be able to afford such expensive and disruptive technologies [14]. Therefore, we speculate that *statistically, organisations' sizes (means they) are significantly different and thus are associated with organisations' adoption behaviours toward BDIDM ($H_{a7}$).*

### 1.3.3. Environmental Factors

Environmental factors that could affect organisations' adoption behaviour include the industry's characteristics (like competition, dominant firms, etc.), the availability of service providers, and whether the regulatory environment, such as government regulations, exists [18]. The industry life cycle can also affect innovation adoption: firms in rapidly growing industries tend to innovate more quickly than those in mature or declining industries. Similarly, the availability of supportive infrastructure, skilled labour, and consultants may affect the adoption [18,43]. Thus, we propose that *industry and market environment have a statistically significant effect on organisations' adoption behaviours toward BDIDM ($H_{a8}$)*. Likewise, *support environment has a statistically significant effect on organisations' adoption behaviours toward BDIDM ($H_{a9}$)*.

Additionally, BDIDM adoption in organisations may be affected by the need to comply with government regulations for protecting users' privacy, like POPIA in South Africa, and some security codes of best practices, like ISO/IEC 27001/2 and NIST [47]. Moreover, the lack of blockchain-specific firm regulations, guidelines, and policies for its adequate standardisation could heavily affect its adoption in organisations [14,36]. For this reason, we posit that *regulatory environment has a statistically significant effect on organisations' adoption behaviours toward BDIDM ($H_{a10}$)*.

The rest of the paper is organised into four sections. Section 2 discusses the methodology followed by executing the research. Section 3 reports the research results, followed by Section 4 discussing the result's implications. The last section, Section 5, concludes the study, highlights major limitations, and provides hints for future research.

## 2. Materials and Methods

This section elaborates on the methodology followed to do this cross-sectional research. The study considered the positivism philosophy, focusing on what could be objectively measured without researchers' interference [48]. It followed a deductive approach since it aimed to test the appropriateness and effectiveness of an existing theory, TOE, for the BDIDM context. Baker [18] claimed that TOE is cross-industry, technology, and context. The study utilised the survey strategy, because it deals with 'what' and 'how' research questions, aligning well with the study's purpose and approach. Awa et al. [43] proved that a survey can be successfully used with the TOE theory. Surveys mitigate the chances for researchers to influence participants by avoiding direct interaction and limiting interference to preserve objectivity [48].

### 2.1. Data Collection

The targeted population to which the study attempts to generalise its results consists of every organisation in the world. Thus, the study's unit of analysis is organisation per se. The research sample consisted of South African organisations, due to the country's proximity to researchers and high exposure to data breaches. Statistics report that Africa has one of the highest rates of cybercrimes and financial losses, with South Africa in the lead in this regard due to its high connectivity rate when compared to the rest of the continent [49].

All qualified participants were targeted. The selection was random, but was limited by the algorithm embedded in the search engine of the medium used (LinkedIn) and restricted to the sample frame. Since identity management is primarily a domain of InfoSec management [1], participants were InfoSec practitioners. However, this population was declared unknown by the Institute of Information Technology Professionals South Africa. The sample frame included different managerial levels of InfoSec to ensure the population representativeness. InfoSec managerial levels are often classified into security officers, managers, and technical staff. Security officers lead digital transformation and set the InfoSec policy in an organisation. They might play a decisive role in adopting BDIDM in their respective enterprises. Security managers are responsible for the security of a specific area of the organisation. They might have distinct perceptions of BDIDM adoption. Technical staff are responsible for implementing, maintaining, and monitoring security

measures according to the InfoSec policy. They might have a pragmatic perspective on BDIDM adoption [1,2].

The data collection instrument was rooted in TOE-BDIDM. The online questionnaire translated every TOE-BDIDM item into a specific close-ended question and was built using Microsoft Forms. Three sections of the questionnaire measured the three main constructs and independent variables of the TOE-BDIDM model, namely: Technology, Organisation, and Environment. The fourth section measured the dependent variable of the TOE-BDIDM model, namely: BDIDM Adoption. The fifth section captured some background information about the sample. Table 1 defines items retained to measure each construct and the independent variable while providing the type of scales used to observe each item.

Data collection took place from February to August 2021. A pilot was conducted for the 15 first records to test the instrument. Requests for participation were regularly sent via LinkedIn direct messages to all 884 prospective participants identified, 15 to 20 participants at a time to allow smooth troubleshooting. This troubleshooting consisted of proving the legitimacy of the research to participants who wished to establish that the participation request was not a disguised cyber-attack. This limitation was expected to constrain the sample size since InfoSec practitioners tend to be ‘reserved’ due to the nature of their jobs [1].

**Table 1.** Primary measurements.

| | Label | Name | Definition | Adapted From | Scale |
|---|---|---|---|---|---|
| Technology | **BDIDM Characteristics:** | **BDIDM_Char** | | **[18] (p. 232)** | **-** |
| | Security (Sec) | Conf | Confidentiality: Resilience to unauthorised view | [1,23,50] | Interval |
| | | Int | Integrity: Resilience to unauthorised change | | |
| | | Avail | Availability: Accessibility to legitimate users when needed | | |
| | Blockchain Type1 | Type1 | Blockchain implementation type | [20] | Nominal |
| | Blockchain Type2 | Type2 | | | |
| | Trialability | Trial | BDIDM easiness of use | [23,41] | Interval |
| | Complexity | Cplex | BDIDM easiness of implementation | [41] | |
| | Observability | Obs | BDIDM easiness of being controlled | [41] | |
| | Compatibility | Cpat | BDIDM easiness of interoperating with other systems | [41] | |
| | Integration | Itegra | BDIDM easiness of smoothly functioning in an ecosystem | [21] | |
| | **BDIDM Readiness:** | **BDIDM_Read** | | **[18] (p. 232)** | **-** |
| | Technology Readiness | Tread | BDIDM preparedness for the enterprise context | [14] (p. 37), [36] (p. 202) | Interval |
| | Standardisation | Std2 | BDIDM normalisation and governance | | |
| | **Infrastructure and Competences:** | **Infr_Comp** | | **[18] (p. 232)** | |
| | Competences | Cpet | Availability of BDIDM competences | [36,43] | Interval |
| | IT Infrastructure | ITInf | Availability of IT infrastructure supportive of BDIDM | [21,35] | |

**Table 1.** *Cont.*

| | Label | Name | Definition | Adapted From | Scale |
|---|---|---|---|---|---|
| **Organisation** | **Organisation Characteristics:** | **Org_Char** | | **[18] (p. 232)** | **-** |
| | Employees Linkage | Net | Formal and informal employees networking supportive of BDIDM | [18] (p. 232) [44] | Interval |
| | Presence Product Champion | Cham | Availability of perceptions of BDIDM value | | |
| | Top management Support | MSup | Strategic support and planning for BDIDM | | |
| | Leadership and Communication | Com | Strategic communication about BDIDM values | | |
| | **Organisation Readiness:** | **Org_Read** | | **[18] (p. 232)** | **-** |
| | Organisation Financial Readiness | ORead | Preparedness for financial investment in BDIDM | [14,18] | Interval |
| | Awareness1 | Awa1 | Awareness of BDIDM | [39] | |
| | Awareness2 | Awa2 | | | |
| | **Organisation Size** | **Size** | **Range of enterprise/Number of employees** | **[18] (p. 232)** | **Nominal** |
| **Environment** | **Support Environment:** | **Sup_Env** | | | **-** |
| | Vendor Support | VSup | BDIDM products vendors support | [18] (p. 232) | Interval |
| | Skill Labour | Slab | BDIDM external skills support | | |
| | Consultants | Cons | BDIDM consultants support | | |
| | **Market and Industry:** | **Ind_Mark** | | **[18] (p. 232)** | |
| | Industry Pressure | Ind | Industry pressure for BDIDM adoption | [18] | Interval |
| | Competition Intensity | Cpeti | BDIDM adoption competition gains | | |
| | **Regulatory Environment:** | **Reg_Env** | | **[18] (p. 232)** | **-** |
| | Government Regulation | Gov | Government pressure for BDIDM adoption | [14,36] | Interval |
| | Compliance with Standards | Std2 | Pressure for BDIDM adoption to comply with standards | [5,47] | |
| | **Adopt Indicator** | **Adopt** | **Adoption intention** | **[18]** | **Binomial** |

### 2.2. Data Analysis

Statistical-paradigm-underlined data analysis was performed using three software applications: Microsoft Excel, the IBM Statistical Package for Social Science (SPSS) version 27.0, and the IBM SPSS analysis of moment structures (AMOS) version 24.0. The study's confidence level was set to 95%, with a subsequent margin of error of 5 percent.

The analysis consisted of three main statistical tests: (i) Structural Equation Modelling (SEM) of the measurement model, known as confirmatory factor analysis (CFA), to test the model fitness and some types of validity and reliability; (ii) binary logistic regression modelling to test the null hypotheses involving variables measured on interval scales and assess the model's predictive accuracy; and (iii) chi-squared tests of goodness-of-fit and association to test null hypotheses involving variables measured on nominal scales. Other important analysis activities included analysing different types of reliability and validity, as well as normality testing and data cleaning.

CFA is often used in social science research [51] to test the fitness of a model with latent variables to particular data. The TOE-BDIDM had latent variables at three levels. The lower level had the security construct; the middle level had BDIDM Characteristics, BDIDM Readiness, Infrastructure and Competencies, BDIDM Characteristics, BDIDM Readiness, Market and Industry, Support Environment, and Regulatory Environment constructs; and the upper level had Technology, Organisation, and Environment constructs. These levelled latent variables lead to a "higher-order CFA" [52] (p. 287) [53] (p. 288). Table 2 describes the fitness indices and their level of acceptance. In the absolute-fit category, Chisq should be insignificant, i.e., with a $p > 0.05$, for a better fit. A significant Chisq indicates a significant difference between the hypothesised model and the actual variance matrices, thus a poorer fit. In addition to testing the model fitness, CFA facilitates the assessment of some reliability

and validity measurements [54]. Table 3 describes the level-of-acceptance of reliability and validity indexes. The internal reliability of scales was estimated using Cronbach's alpha and parallel forms techniques.

A binary logistic regression modelling "aims to see whether a value of the binary dependent variable can be predicted by the score of an independent variable" [55] (p. 319). In this study, the test sought to see if scores in the TOE factors, i.e., the eight middle latent variables, could predict adopters ('1' value) and non-adopters ('0' value) in the Adopt Indicator dependent variable. Binary logistic regression analysis was more suitable than path analysis mainly because the dependent variable's scale is dichotomous. Path analysis involves linear regression [56] and requires that the dependent variable be measured on a ratio [57] or interval scale [58].

**Table 2.** Fitness indexes acceptance level.

| Name of Category | Name of Index | Level of Acceptance | Reference |
|---|---|---|---|
| Absolute Fit | Discrepancy chi-square (Chisq) | $p > 0.05$ | [59] |
| | Root Mean Square of Error Approximation (RMSEA) | RMSEA < 0.08 | [60] |
| Incremental Fit | Goodness-of-Fit Index (GFI) | GFI > 0.90 | [61] |
| | Adjusted Goodness-of-Fit (AGFI) | AGFI > 0.90 | [62] |
| | Comparative Fit Index (CFI) | CFI > 0.90 | [63] |
| | Tucker–Lewis Index (TLI) | TLI > 0.90 | [64] |
| | Normed Fit Index (NFI) | NFI > 0.90 | [65] |
| Parsimonious Fit | Chi-squared/Degree of freedom (Chisq/df) | Chisq/df < 5.0 | [66] |

**Table 3.** Reliability and validity indexes acceptance level.

| Name of Category | Name of Index | Level of Acceptance | Reference |
|---|---|---|---|
| Internal reliability | Cronbach's alpha ($\alpha$) | $\alpha \geq 0.5$ | [67] |
| Construct reliability | Composite reliability (CR) | $CR \geq 0.6$ | [67] |
| Convergent Validity | Average Variance Extracted (AVE) | $AVE \geq 0.5$ | [67] |
| Discriminant Validity (multicollinearity) | Tolerance (Tol.) or Variance Inflation Factor (VIF) | Tolerance > 0.1 | [68] |
| | | VIF < 10 | [54] |

In IBM Amos SPSS, the binary logistic regression test involves an omnibus test, Hosmer–Lemeshow goodness-of-fit test, and Cox–Snell R-Squared ($R^2$) values to estimate the explanatory strength of the latent variables. The significance of the regression depends on the significance of the omnibus test ($p < 0.05$) and the insignificance of the Hosmer–Lemeshow test ($p > 0.05$). Wald statistics also accompany the test to determine the significance of each factor of the logistic regression. A classification table displays the number of observed cases the model predicted accurately and their percentages [55,58]. This test was appropriate for all the null hypotheses except $H_{02}$ and $H_{07}$.

$H_{02}$ and $H_{07}$ involve nominal variables, which were tested separately using the chi-squared test of goodness-of-fit and association. The chi-squared test of goodness-of-fit compares expected and observed frequencies of the categories of a variable to assess whether there is a significant difference between them [58]. The chi-squared test of association seeks to "compare two different sets of frequency counts to see" if they are associated [55] (p. 13).

## 3. Results

This section reports the study's findings based on statistical tests performed on data as predesigned. The survey recorded 115 responses out of 884 requests sent, indicating a participation rate of 13%. The survey was set to only record valid responses to anticipate data cleaning. Thus, no missing data were found in the dataset. However, four records

were excluded because they were beyond the sample frame or ethics requirements: two respondents were outside South African organisations, and two were younger than 18. This led to a sample size of 111. This sample size is arguably acceptable given the good internal consistency and exceptional "missing data level" of 0, though it led to 4.8 responses per parameter, close to the minimum requirement of five responses per parameter [69] (p. 2223). The distributions were visualised in boxplots to indicate the five extreme values as outliers. Records with outliers were not deleted to avoid unnecessary shrinking of the sample size. Except for nominal variables, all outliers were replaced by the average values of the respective distributions.

### 3.1. Demographic Information

The 111 InfoSec practitioners had various backgrounds, but their profiles aligned well with the sample frame as shown in Table 4: 14 officers, 16 managers, and 78 technical staff. The executive level was not anticipated but aligned well with the targeted profiles. Respondents could enter their job titles when they could not pick one which was representative of their position from the list provided. The table also shows that all four organisation sizes were represented in the sample. The organisation's sizes were determined by the number of employees as estimated by each respondent: above 250 employees for large enterprises, 51 to 250 employees for medium enterprises, 11 to 50 employees for small enterprises, and below 10 employees for micro-enterprises. Of the 111 respondents, 70 belonged to large enterprises and 41 belonged to SMEs. The pie chart in Figure 2 describes organisation sectors represented in the sample according to the Statistics South Africa department classification (stats-sa, inedi). Most of the respondents' organisations fall under information and communication technology (47 respondents, accounting for 42.3 % of the sample) or financial and insurance (30 respondents, accounting for 27 % of the sample) sectors. Table 5 gives additional background information, including age group, IDM system existence, awareness of BDIDM, blockchain type, and the intention to adopt BDIDM.

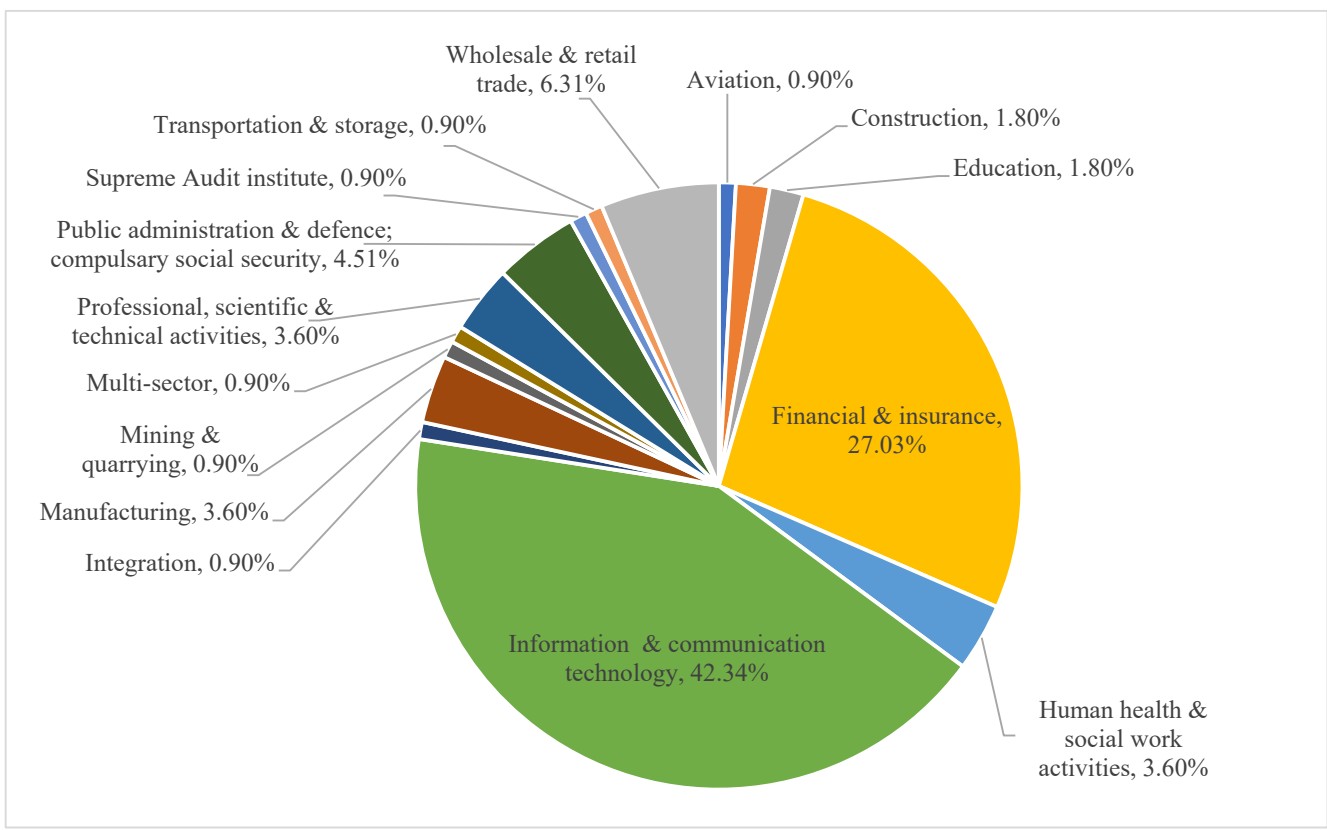

**Figure 2.** The sample by organisation sector.

**Table 4.** Sample managerial levels.

| InfoSec Managerial Levels | | | | Organisation Size | | | |
| --- | --- | --- | --- | --- | --- | --- | --- |
| Job Title | Freq. Job | Freq. Level | Percent Level | Micro | Small | Medium | Large |
| **Executive:** | | **3** | **2.70** | **0** | **1** | **0** | **2** |
| InfoSec Governance/Executive | 3 | | | | | | |
| **Officer:** | | | | | | | |
| IAM Officer | 7 | | | | | | |
| Chief Information Security Officer | 3 | **14** | **12.61** | **2** | **3** | **1** | **8** |
| Chief Information Officer | 2 | | | | | | |
| Chief Technology Officer | 2 | | | | | | |
| **Manager:** | | | | | | | |
| InfoSec Manager | 11 | | | | | | |
| InfoSec Architecture Manager | 1 | | | | | | |
| Network Security Manager | 1 | **16** | **14.41** | **1** | **1** | **1** | **13** |
| Data Centre Manager | 1 | | | | | | |
| Operation Manager | 1 | | | | | | |
| Project Manager | 1 | | | | | | |
| **Technical Staff:** | | | | | | | |
| InfoSec Administrator/ Analyst/Specialist/Architect/Consultant | 40 | | | | | | |
| IAM Administrator/ Analyst/Specialist/Consultant/Engineer | 9 | | | | | | |
| Network Security Administrator | 5 | | | | | | |
| IT auditor/Program Analyst | 3 | | | | | | |
| Cloud Administrator/ Engineer/Consultant | 4 | **78** | **70.27** | **3** | **8** | **20** | **47** |
| System Administrator/Engineer | 4 | | | | | | |
| Software Developer/Engineer | 2 | | | | | | |
| Solutions Architect | 4 | | | | | | |
| Data Engineer | 2 | | | | | | |
| PenTester, ERP Analyst, Technical Support, Technical Engineer | 5 | | | | | | |
| **Total** | **111** | **111** | **100.00** | **6** | **13** | **22** | **70** |
| | | | **Total** | | **111** | | |

**Table 5.** Additional demographic information.

| Question | Answer | Frequency | Percent |
|---|---|---|---|
| Age group | Between 18 and 40 | 82 | 71.0 |
| | Over 40 | 31 | 27.0 |
| Does your organisation have an established IDM system? | I don't know | 3 | 2.7 |
| | No | 8 | 7.2 |
| | Yes | 100 | 90.1 |
| | Total | 111 | 100.0 |
| Is your organisation aware of BDIDM? | No | 40 | 36.0 |
| | Not sure | 18 | 16.2 |
| | Yes | 53 | 47.7 |
| | Total | 111 | 100.0 |
| Which type of blockchain do you think is suitable for your organisation's BDIDM? | Public permissionless blockchain | 10 | 9.0 |
| | Public permissioned blockchain | 27 | 24.3 |
| | Private permissioned blockchain | 74 | 66.7 |
| | Total | 111 | 100.0 |
| Would you recommend BDIDM to your organisation? | Yes | 80 | 72.1 |
| | No | 31 | 27.9 |
| | **Total** | **111** | **100.0** |

### 3.2. Assessing Internal and Parallel-Forms Reliabilities

In Table 6 column 'α' shows that all constructs had good internal consistency. A good percentage of the variance observed in the constructs is accurate and reliable: 78% for security, 72% for BDIDM Characteristics, 65% for BDIDM Readiness, 69% for IT Infrastructure and Competences, 78% for Organisation Characteristics, 83% for Organisation Readiness, 78% for Support Environment, 89% for Industry and Market Environment, and 78% for Regulatory Environment. The security variable part of BDIDM Characteristics was computed as the average of its constituents as predesigned according to the CIA triad theory. The Cronbach's alpha test suggested that Awareness1 be excluded from the Organisation Characteristics construct to improve its internal consistency to 0.83 from 0.45. However, before excluding it, the test of the parallel form was performed to assess any possibility of combining it with Awareness2.

**Table 6.** Internal reliabiity.

| | Cronbach's Alpha | N of Items |
|---|---|---|
| Sec | 0.78 | 3 |
| BDIDM_Char | 0.72 | 6 |
| BDIDM_Read | 0.65 | 2 |
| Infr_Comp | 0.69 | 2 |
| Org_Char | 0.78 | 4 |
| Org_Read | 0.83 | 2 |
| Sup_Env | 0.78 | 3 |
| Reg_Env | 0.78 | 2 |
| Ind_Mark | 0.89 | 2 |

The questionnaire had some parallel questions (i.e., different questions measuring the same items) set intentionally to anticipate internal reliability. The parallel-form reliability estimates the correlation between such variables to test the consistency of answers [70]. The Spearman's correlation test showed that Blockchain Type1 and Blockchain Type2 correlated

absolutely since the *p*-value was 0.0. Thus, the study's observation about Blockchain Type for the enterprise context is accurate and reliable. Blockchain Type2 was chosen for the rest of the analysis because it was more reliable. The Pearson correlation test shows that Awareness1 and Awareness2 were not correlated, as the *p*-value was 0.06. The survey questions might not have been semantically identical enough and thus measured distinct aspects of awareness. Thus, Awareness1 and Awareness2 could not be combined. Awareness2 was chosen over Awareness1 because it was far more reliable.

### 3.3. SEM of the Measurement Model—CFA

The higher-order CFA shown in Figure 3 represents TOE-BDIDM's measuring model performed on data using IBM Amos SPSS. Directional arrows indicate causal relationships, bidirectional arrows indicate covariance relationships, and circles measure error on each 'caused' variable. Numbers on directional arrows indicate the factor loadings and those on bidirectional arrows indicate covariance rate between constructs. Rectangles represent indicators or items. Ovals in the middle represent first-order constructs (the eight latent variables), while those on the right-hand side represent second-order constructs (the three contexts of the TOE model). It is important to recall that Organisation Size and Blockchain Type items are excluded from CFA because they are nominal scales. Table 7 reports the fitness indices for the hypotheses' BDIDM model as illustrated in Figure 3 and the modified TOE-BDIDM model.

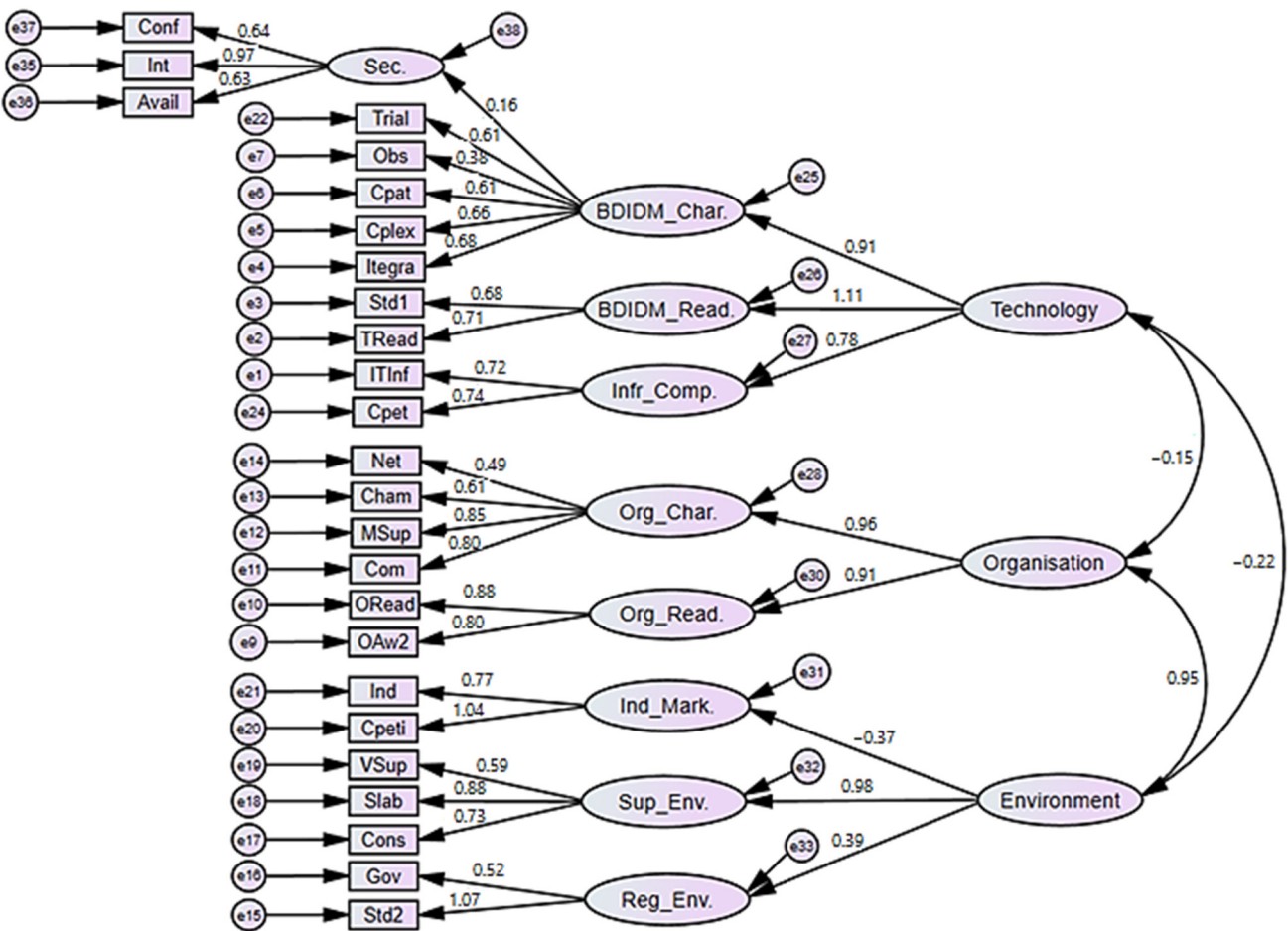

**Figure 3.** Higher-order CFA of the hypothesised TOE-BDIDM model.

**Table 7.** Fitness indices results.

|  | Hypothesised Model | Modified Model |
|---|---|---|
| *p* | 0.0 | 0.14 |
| RMSEA | 0.08 | 0.03 |
| GFI | 0.77 | 0.87 |
| AGFI | 0.71 | 0.82 |
| CFI | 0.84 | 0.98 |
| TLI | 0.82 | 0.98 |
| NFI | 0.70 | 0.86 |
| Chisq/df | 01.7 | 1.1 |

Fitness indices of the hypothesised model suggested a poor fit for this data. The model was modified to improve the fit by (i) excluding items with a poor factor loading of $\lambda < 0.5$, (ii) adding some covariances between errors as suggested by improvement indices of the test, and (iii) solving a suspected 'redundancy' signalled by a very high covariance of 0.95 between Technology and Environment constructs. The Environment construct appeared more problematic because only one of its constituents, Sup_Env, was strongly loaded with $\lambda = 0.98$. The rest were unusually weak. The three constituents (Ind_Mark, Sup_Env, and Reg_Env) do not share a fair portion of variance with the Environment construct they intended to measure. Unexpectedly, Sup_Env was also strongly loading on the Organisation construct with $\lambda = 0.95$, causing it to be redundant with Environment. The problem was solved by moving Sup_Env into the Organisation construct and isolating the remaining variables.

Nevertheless, this change did not affect further binary logistic regression analysis since it only involved the first-order constructs. The suggestion of solving redundancy by either "combining the highly correlated variables through principal component analysis, or omitting a variable from the analysis that associated with another variable (s) highly" [68] (p. 5) was not applicable. The redundancy happened at the higher level of the model, making it unreasonable to combine or omit such constructs. Moreover, the collinearity test performed earlier using linear regression analysis in SPSS suggested no case of multicollinearity. Table 8 shows that all items' and constructs' tolerance values were greater than 0.1, and their respective VIF values less than 10. Therefore, the items and first-order constructs of the hypothesised model are distinctive, fulfilling the requirement of discriminant validity.

Table 8 reports good construct reliability of the hypothesised model as nearly all of the CR values for both first- and second-order constructs were greater than 0.6, except for the Technology construct which was entirely below this threshold. This result means that an overall good proportion of variance is shared between indicators and the constructs they measured.

Of the nine first-order constructs, Table 8 shows that eight did more-or-less meet the acceptable level of convergent validity as their AVE values are greater than 0.5, even though BDIDM_Read and Org-Char's values are just about that threshold while BDIDM_Char's is remarkably low. BDIDM-Char's AVE value did not improve after excluding poorly loading factors. The modified model reported AVE values of 0.39 for BDIDM_Char, 0.49 for BDIDM_Read, and 56 for Org-Char. Therefore, Sec and Obs items were reconsidered in the binary logistic modelling. This choice was also motivated by BDIDM_Char's excellent internal consistency of $\alpha = 0.72$ and good construct reliability of CR = 0.68 in the hypothesised model.

**Table 8.** Summary of Reliability and validity indexes.

| Item | | | | First-Order Construct | | | | | | | Second-Order Construct | | |
|------|------|------|------|------|------|------|------|------|------|------|------|------|------|
| **Name** | **Tol0.** | **VIF** | **κ** | **Name** | **CR** | **AVE** | **α** | **VIF** | **Tol0.** | **κ** | **Name** | **CR** | **AVE** |
| Conf | 0.41 | 2.43 | 0.64 | | | | | | | | | | |
| Int | 0.32 | 3.09 | 0.97 | Sec | 0.80 | 0.58 | 0.78 | | | | | | |
| Avail | 0.45 | 2.24 | 0.63 | | | | | | | | | | |
| Sec | | | 0.16 | | | | | | | | | | |
| Obs | 0.51 | 1.96 | 0.38 | | | | | | | | | | |
| Itegra | 0.36 | 2.76 | 0.73 | | | | | | | | | | |
| Cplex | 0.48 | 2.10 | 0.61 | BDIDM_Char | 0.68 | 0.29 | 0.72 | 0.46 | 2.17 | 0.91 | Technology | 0.75 | 0.11 |
| Cpat | 0.41 | 2.43 | 0.56 | | | | | | | | | | |
| Trial | 0.55 | 1.82 | 0.58 | | | | | | | | | | |
| Std1 | 0.42 | 2.37 | 0.66 | BDIDM_Read | 0.65 | 0.49 | 0.65 | 0.38 | 2.64 | 10.11 | | | |
| TRead | 0.39 | 2.54 | 0.73 | | | | | | | | | | |
| ITInf | 0.39 | 2.55 | 0.66 | Infr_Comp | 0.69 | 0.53 | 0.69 | 0.59 | 1.72 | 0.78 | | | |
| Cpet | 0.44 | 2.27 | 0.79 | | | | | | | | | | |
| Net | 0.44 | 2.18 | 0.49 | | | | | | | | | | |
| Cham | 0.44 | 2.29 | 0.57 | Org_Char | 0.78 | 0.48 | 0.78 | 0.42 | 2.40 | 0.96 | | | |
| Com | 0.33 | 3.05 | 0.79 | | | | | | | | Organisation | 0.65 | 0.13 |
| MSup | 0.26 | 3.92 | 0.85 | | | | | | | | | | |
| ORead | 0.28 | 3.57 | 0.86 | Org_Read | 0.82 | 0.70 | 0.83 | 0.42 | 2.40 | 0.91 | | | |
| OAw2 | 0.39 | 2.59 | 0.80 | | | | | | | | | | |
| Cons | 0.39 | 2.58 | 0.73 | | | | | | | | | | |
| Slab | 0.28 | 3.62 | 0.89 | Sup_Env | 0.78 | 0.55 | 0.78 | 0.59 | 1.71 | 0.98 | | | |
| VSup | 0.57 | 1.74 | 0.58 | | | | | | | | Environment | 0.45 | 0.58 |
| Std2 | 0.45 | 2.22 | 0.83 | Reg_Env | 0.72 | 0.57 | 0.78 | 0.39 | 2.57 | 0.39 | | | |
| Gov | 0.46 | 2.19 | 0.67 | | | | | | | | | | |
| Ind | 0.25 | 4.01 | 0.88 | Ind_Mark | 0.79 | 0.80 | 0.89 | 0.59 | 1.71 | −0.37 | | | |
| Cpeti | 0.22 | 4.50 | 0.91 | | | | | | | | | | |

Of the three second-order constructs, only the Environment's AVE value met the threshold of greater than 0.6. The modification done on the model did not improve this result, as the Technology and Organisation's AVEs were still below 0.6, moving from 0.11 to 0.12 and from 0.13 to 12, respectively. However, the CFA indicated that nearly all regression weights of the hypothesised model were statistically significant. Therefore, the convergent validity of the hypothesised model is arguably acceptable. In other words, the variation in the constructs is reasonably explained by their respective item constituents. However, the hypothesised model did not fully meet the requirement concerning construct validity since the CFA suggested that it did not perfectly fit the data. The modified model offered a better fit.

### 3.4. Normality Testing

Any regression analysis requires data to be normally distributed. Normality testing only relied on boxplot visualisation, skewness, and kurtosis [71]. The Shapiro and Kolmogorov tests are often difficult to pass for a sample size below 300 [72]. A common rule in skewness and kurtosis assessment suggests that the statistic's ratio on the test's standard error should be less than the Z-distribution's critical value of 1.96 [73]. Another rule adds that the skewness statistic should be less than 0.8 [74]. The tests performed on the eight distributions revealed that only Org_Read and Reg_Env had their skewness critical value greater than 1.96. However, their skewness and kurtosis statistics were mostly less than 0.8. Therefore, the eight distributions are approximately normally distributed and fit further analysis.

### 3.5. Binary Logistic Regression Analysis

Binary logistic regression analysis considered the hypothesis model over the modified since it has no redundant items or first-order constructs. The study's hypotheses concern the direct relationship between the eight latent variables and the dependent variable without any moderation. The latent variables were computed as the average of their respective items. The binary logistic regression modelling was performed using IBM Amos SPSS and utilised the 'Enter' method. The enter method means the factors were simultaneously inserted together in the regression model in Step 1 to see their overall interaction. No factor was delayed or privileged over others.

The test showed a significant omnibus result of $\chi^2$ = 31.15, df = 8, $p$ = 0.0 and an insignificant Hosmer result of $\chi^2$ = 8.48, df = 8, $p$ = 0.39. These results indicate that the binary logistic regression was significant. The variance observed in the Adopt Indicator dependent variable was due to the variance observed in the TOE factors (the eight latent variables), rather than randomness. Therefore, there is a relationship between the TOE factors and organisations' adoption behaviours toward BDIDM. The test reported a Cox–Snell $R^2$ of 0.245 to 0.351. Since the study's confidence level is 95%, this interval meant that if the data collection was repeated 20 times, 19 would have 24.5 to 35.1% of the variance observed in the Adopt Indicator due to the variance in the TOE factors. The classification table in Block 1 (see Table 9) shows that the hypothesised model has an overall predictive accuracy of 79.3%, a higher percentage than that of the basic model in Block 0, which displayed 72.1 percent. The hypothesised model was exceptionally accurate in predicting BDIDM adopters, more so than non-adopters: 92.5% of adopters were accurately predicted compared to only 45.2% of non-adopters accurately predicted.

**Table 9.** Classification table.

| | | | | | | Classification Table [a] | | | |
|---|---|---|---|---|---|
| | | | | | Predicted | | |
| | **Observed** | | | Adopt Indicator | | Percentage Correct |
| | | | | Yes | No | |
| Step 1 | Adopt Indicator | Yes | 74 | 6 | 92.5 |
| | | No | 17 | 14 | 45.2 |
| | Overall Percentage | | | | 79.3 |

[a]. The cut value is 0.500.

Wald statistics in Table 10 show the significance of each of the eight factors included in the regression, with the values of the 'B' column representing the values each could predict. A Yes value is predicted with positive values and a No value with negative values. It can be seen that only Org_Read factors could predict a 'Yes' in the Adopt Indicator. Thus, it has a positive effect on BDIDM adoption. The rest of the factors could predict a 'No' value in the Adopt Indicator. Thus, they have a negative effect on BDIDM adoption.

However, the column 'Sig' of the table reveals that only BDIDM_Char's effect is statistically significant since it is the only factor displaying a $p$-value of less than 0.05 (Wald = 5.415, df = 1, Sig. = 0.02). Therefore, BDIDM characteristics, which is made up of Security, Trialability, Complexity, Observability, Compatibility, and Integration items, constituted the most significant factor that negatively affects the likelihood of adopting BDIDM in an organisation. The data provided enough evidence supporting that the more BDIDM is insecure, uncontrollable, user-unfriendly, complex, incompatible with other systems, and challenging to integrate into the enterprise ecosystem, the less likely an organisation would be to adopt it.

As a result, of the eight underlying null hypotheses, only $H_{01}$ was rejected, which confirmed the study's alternative hypothesis $H_{a1}$, stating that BDIDM Characteristics have a statistically significant effect on BDIDM Adoption.

**Table 10.** Wald statistics.

| | | B | S.E. | Wald | df | Sig. | Exp(B) | 95% C.I. for EXP(B) | |
|---|---|---|---|---|---|---|---|---|---|
| | | | | | | | | Lower | Upper |
| | BDIDM_Char | −1.640 | 0.705 | 5.415 | 1 | 0.020 | 0.194 | 0.049 | 0.772 |
| | BDIDM_Read | −0.167 | 0.442 | 0.142 | 1 | 0.706 | 0.847 | 0.356 | 2.012 |
| | Inf_Comp | −0.318 | 0.347 | 0.837 | 1 | 0.360 | 0.728 | 0.368 | 1.438 |
| | Org_Char | −0.312 | 0.672 | 0.215 | 1 | 0.643 | 0.732 | 0.196 | 2.734 |
| Step 1 [a] | Org_Read | 1.316 | 0.691 | 3.626 | 1 | 0.057 | 3.730 | 0.962 | 14.458 |
| | Indu_Mark | −0.280 | 0.349 | 0.643 | 1 | 0.422 | 0.756 | 0.381 | 1.498 |
| | Sup_Env | −0.707 | 0.725 | 0.951 | 1 | 0.329 | 0.493 | 0.119 | 2.041 |
| | Reg_Env | −0.494 | 0.354 | 1.948 | 1 | 0.163 | 0.610 | 0.305 | 1.221 |
| | Constant | 6.258 | 2.920 | 4.593 | 1 | 0.032 | 522.069 | | 0.772 |

[a]. Variable(s) entered on step 1: BDIDM_Char, BDIDM_Read, Inf_Comp, Org_Char, Com_Proc, Org_Read, Indu_Mark, Sup_Env, Reg_Env.

### 3.6. Chi-Square Tests of Goodness of Fit and Association

Chi-squared tests were intended to test $H_{02}$ and $H_{07}$, as these were not part of the above logistic regression due to their nominal nature. This test was done in two steps: Chi-squared test of goodness-of-fit and Chi-squared test of association. The chi-squared test of goodness assessed the significance of the difference between categories for both Blockchain Type and Organisation Size variables, represented by $H_{02\cdot1}$ and $H_{07\cdot1}$, respectively. The test reported a *p*-value of less than 0.05 for both Blockchain Type and Organisation Size variables, suggesting that the categories were indeed different. The chi-squared test of association assessed whether the outcome in the Adopt Indicator dependent variable was associated with the categories for both Blockchain Type and Organisation Type variables, represented by $H_{02\cdot2}$ and $H_{07\cdot2}$, respectively. As shown in Table 11, the test reported *p*-values of more than 0.05 for Organisation Size and less than 0.05 for Blockchain Type variables. These results suggest that only the Blockchain Types are associated with BDIDM adoption. Therefore, an organisation can decide whether to adopt BDIDM or not to based on the blockchain types involved. Table 12 summarises the outcome of all hypothesis-testing activities. The study failed to reject the null hypotheses $H_{03}$, $H_{04}$, $H_{05}$, $H_{06}$, $H_{08}$, $H_{09}$, and $H_{010}$. This means there is not enough statistical evidence to support $H_{a3}$, $H_{a4}$, $H_{a5}$, $H_{a6}$, $H_{a8}$, $H_{a9}$, and $H_{a10}$. Only $H_{01}$ and $H_{02}$ were rejected, which confirmed the study's alternative hypothesis $H_{a1\ and}\ H_{a2}$.

**Table 11.** Chi-squared test of association—Significance.

| Chi-Square Tests | | | |
|---|---|---|---|
| **Organisation Size** | **Value** | **df** | **Asymptotic Significance (2-Sided)** |
| Pearson Chi-Squared | 4.268 [a] | 3 | 0.234 |
| Likelihood Ratio | 5.960 | 3 | 0.114 |
| Linear-by-Linear Association | 2.295 | 1 | 0.130 |
| N of Valid Cases | 111 | | |
| **Blockchain Type2** | **Value** | **df** | **Asymptotic Significance (2-Sided)** |
| Pearson Chi-Squared | 6.863 [b] | 2 | 0.032 |
| Likelihood Ratio | 8.692 | 2 | 0.013 |
| Linear-by-Linear Association | 6.653 | 1 | 0.010 |
| N of Valid Cases | 111 | | |

[a] 3 cells (37.5%) have expected count less than 5. The minimum expected count is 1.68. [b] 1 cells (16.7%) have expected count less than 5. The minimum expected count is 2.79.

**Table 12.** Summary of hypothesis testing.

| | Null Hypothesis Tested | | Outcome |
|---|---|---|---|
| **Technology** | $H_{01}$: BDIDM Characteristics do not have a statistically significant effect on organisations' adoption behaviours toward BDIDM. | | R |
| | $H_{02}$: $H_{02.1}$: Statistically, Blockchain Types are equal. | R | R |
| | $H_{02.2}$: Statistically, Blockchain Types are not associated with organisations' adoption behaviours toward BDIDM. | R | |
| | $H_{03}$: BDIDM Readiness does not have a statistically significant effect on organisations' adoption behaviours toward BDIDM. | | FR |
| | $H_{04}$: IT Infrastructure and Competencies do not have a statistically significant effect on organisations' adoption behaviours toward BDIDM. | | FR |
| **Organisation** | $H_{05}$: Organisation Characteristics do not have a statistically significant effect on BDIDM Adoption. | | FR |
| | $H_{06}$: Organisation Readiness does not have a statistically significant positive effect on BDIDM Adoption. | | FR |
| | $H_{07}$: $H_{07.1}$: Statistically, Organisation Sizes are equal. | R | FR |
| | $H_{07.2}$: Statistically, Organisation Sizes are not associated with organisations' adoption behaviours toward BDIDM. | FR | |
| **Environement** | $H_{08}$: Industry and Market Environment do not have a statistically significant effect on organisations' adoption behaviours toward BDIDM. | | FR |
| | $H_{09}$: Support Environment does not have a statistically significant effect on organisations' adoption behaviours toward BDIDM. | | FR |
| | $H_{010}$: Regulatory Environment does not have a statistically significant effect on organisations' adoption behaviours toward BDIDM. | | FR |

Note: R = Rejected, FR = Failed to Reject.

### 3.7. Summary of Findings

The binary logistic regression analysis confirmed the relationship between the TOE factors and BDIDM adoption decision making. The factors predict whether an organisation would fall under the BDIDM Adopters or Non-Adopters category. The effect of each of the TOE factors was either positive or negative. A positive predictor suggests that positive growth in its score results in an increased likelihood of an organisation adopting BDIDM. A negative predictor suggests that negative growth in its score results in a decreased likelihood of an organisation adopting BDIDM. BDIDM Characteristics happened to be the only statistically significant factor in the regression.

The Chi-squared tests, on one hand, confirmed the notion that blockchain types are associated with organisations' adoption behaviour toward BDIDM and, on the other hand, refuted the notion that organisation sizes are not associated with the behaviour.

SEM of the measurement model reveals that TOE-BDIDM is highly effective in predicting adopters of BDIDM, more so than non-adopters, accurately predicting 92.5% of adopters but only 45.2% of non-adopters. TOE-BDIDM tends to be faulty on construct validity since it did not perfectly fit the data. However, the model displays excellent internal reliability, good construct reliability, and arguably reasonable convergent and discriminant validity. Hence, the general view is that the model is relatively appropriate for the context of enterprise BDIDM adoption.

## 4. Discussion

This section discusses the implications of the findings and their significance from the literature perspective, considering the research objectives, practice, and theory.

### 4.1. Implications for the Study's Objectives

The study examined the effect of TOE factors on organisations' adoption behaviours toward BDIDM, anticipating a relationship between them. The findings confirmed the

relationship, supporting that TOE factors do predict how an organisation "identifies the need, searches, and adopts new technologies" [18] (p. 232). Typically, the factors either promote or constrain the adoption [14], predicting BDIDM adopters and non-adopters.

This study also argued that some TOE factors were more critical than others, assuming they were statistically significant, and affected the adoption behaviour in organisations the most. The data suggests that this argument, alongside the assumption, is correct. The most critical factor happened to be BDIDM characteristics. This confirms the literature's view that the disruptiveness of BDIDM heavily affects its adoption in organisations [6]. Another result that verified the significance of some TOE factors over others was the statistical significance of the association of the Blockchain Type item with the adoption behaviour in organisations. Blockchain Type item was designed as part of the BDIDM Characteristics factor, which was already statistically significant in the regression model. This implies some degree of consistency in the measurements and validity of the results.

The reasonable appropriateness and effectiveness of TOE-BDIDM means that these findings are mostly rational and consistent. TOE-BDIDM was excellent in predicting BDIDM adopters at an accuracy rate of 92.5 percent. The model tends to be less effective in predicting BDIDM non-adopters at an accuracy rate of 45.2 percent. The TOE-BDIDM's overall predictive accuracy sits at 79.3 percent. These results are valid since Awa et al.'s [43] TOE-based model displayed a similar pattern, with about the same overall predictive accuracy of 78.7%, but at different individual proportions of 87.1% for ERP adopters and 66.7% for ERP non-adopters. The CFA suggested a poor fit of the hypothesised TOE-BDIDM. The modified model offered a better fit. However, most of the items of the hypothesised model were loading well on the constructs, indicating fair composite reliability. Nearly the entire model had an excellent internal consistency. Therefore, TOE-BDIDM as hypothesised is relatively appropriate for the BDIDM context. TOE-BDIDM identified two significant relationships with N = 111, at $p < 0.05$ compared to three identified in a similar study [43] with N = 373, at $p < 0.05$. This verifies the assumption of the TOE theory's ability to explain the enterprise adoption phenomenon in the contex of IDM and validates TOE's interoperability [18].

### 4.2. Implications for Practice

BDIDM Characteristics is the only statistically significant factor in the regression in contrast to Awa et al. [43]'s study that found that Organisation Size and Support Environment were statistically significant at $p < 0.05$, N = 373. The insignificance of Support Environment in the present study may mean that outsourcing BDIDM solutions is perhaps not a sustainable solution for organisations, due to privacy issues and the legal responsibilities involved [16,75]. The results also contrast against the literature's suggestion that SMEs may be less likely to adopt BDIDM than large enterprises [45] due to its relatively high cost of implementation [14]. This means BDIDM disruptiveness could be leveraged by any organisation regardless of its size. This finding may be encouraging to SMEs with little or no preexisting IDM systems. They could leverage the lower cost of rebooting their IT infrastructure into a distributed architecture [21,35].

Critics have highlighted the principle vulnerability of BDIDM and SSI dwelling at its endpoints [31], questioning whether further adoption ought to be encouraged [33]. From this perspective, the significance of BDIDM Characteristics may imply that more investment will be directed toward making BDIDM more secure, controllable, user-friendly, less complex to implement, more compatible with other systems, and smoother to integrate into the enterprise ecosystem. However, some of these goals might be extremely difficult to achieve because blockchain was somehow designed to function as such. With the imminence of ZT security architecture [26], perhaps the question should shift from 'how to make BDIDM organisation-friendly' to 'how to make an organisation BDIDM-friendly'. "Reluctance to adopt disruptive technologies might be a significant competitive disadvantage for an organisation, whereas proactive planning can be a significant advantage" [14] (p. 34).

The significance of the Blockchain Type item echoes the debate around the ideal blockchain implementation for organisations [25]. Adopters preferred the private permissioned blockchain type, supporting the view that this blockchain is more suitable for the enterprise context [21]. They indirectly disapprove of SSI, holding the view that it is the typical BDIDM model fitting into a public permissioned blockchain [16]. This finding may suggest that SSI is not yet fully sustainable for the enterprise context. The SSI value proposition for organisations is not well understood or is yet to be adequately articulated [18].

From a practical significance perspective, Organisation Readiness (involving Financial Readiness and Organisation Awareness items) is the second critical factor. It was the least statistically insignificant, with a *p*-value of 0.57, just above the threshold of 0.5. Moreover, this factor is as expected and the only to have a positive effect on BDIDM adoption, thus, the only factor predicting BDIDM adopters. The criticality of Organisation Readiness echoes the so-often-highlighted effect of financial resources in adopting blockchain-based solutions [14]. It also reflects the importance of awareness of BDIDM in promoting its adoption due to its relative newness [39].

*4.3. Implications for Theory*

The criticality of the BDIDM Characteristics factor, including the Blockchain Type item, suggests that organisations' adoption behaviour toward BDIDM is "more driven by technological factors than by organisational and environmental factors" [43] (p. 16), just like Awa's study in the context of ERP adoption in SMEs. The criticality of BDIDM Characteristics, broadly relating to the Technology Characteristics of the TOE theory, resonates with Baker's view portraying BDIDM as a disruptive and radical innovation, producing a discontinuous change intended to shift the InfoSec paradigm [6]. Although its proper implementation is costly, requiring a highly skilled team [14], its cost savings in password management alone is estimated in the millions of dollars [34]. Therefore, it could be theorised that BDIDM has both a high risk related to its adoption and the potential to "enhance competitive standing in an organisation" [18] (p. 232). BDIDM-derived competitive advantages include enhancing a sustainable IDM through the maintaining of privacy.

The TOE theory's flexibility, which allows for customisation to different constructs [18], tends to leave room for ambiguities. The Support Environment factor is referred to by some literature as "Technical Know-how" [43] (p. 7), locating it under the Technology construct, while others [18] identify it as external support, locating it under the Environment construct. Contrary to both views, this study suggests that it fits the Organisation construct well, which led to the modification of the model. Moreover, the TOE theory should extend to include a 'User construct' to capture a more comprehensive context of BDIDM adoption in organisations. In contrast to traditional IDM systems, SSI is user-centric [76] due to identity self-management preserving privacy [15,16]. This "could practically introduce novel issues" [7] (p. 106). The BDIDM system would rely heavily on users to "safeguard against forgetting (or losing) the private key" [32] (p. 5). Therefore, a complete explanation of organisations' adoption behaviours toward BDIDM might necessitate measuring individual factors, such as User Preparedness, Willingness, Acceptance, Skills, Awareness, Perceived Usefulness, etc.

## 5. Conclusions

This study primarily sought to explain the predictive effect of TOE factors on organisations' adoption behaviours toward BDIDM to determine the most critical predictors and examine their effect. Technology Characteristics happened to be the most critical factor and predicted BDIDM non-adopters the most. The more BDIDM is insecure, uncontrollable, user-unfriendly, complex, incompatible with other systems, and challenging to integrate into the enterprise ecosystem, the less likely an organisation would be to adopt it. Blockchain Type, which is an item within the Technology Characteristics construct, was confirmed to be associated with organisations' adoption behaviour, supporting the notion

that an organisation can adopt BDIDM because of the type of blockchain involved. The association between organisation sizes and BDIDM adoption happened to be statistically insignificant, supporting that SMEs are as likely to adopt BDIDM as larger organisations.

The majority of respondents intended to recommend BDIDM to their organisations yet, paradoxically, preferred private permissioned blockchain type the most, revealing resistance to decentralised and privacy-preserving BDIDM models like SSI. The latter might be utopian or impractical for organisations. Consequently, blockchain technology might not necessarily dissolve intermediation in IDM in organisations but rather transform it to maintain centralisation.

Regarding the TOE performance, the general view is that the TOE model is relatively appropriate for the context of enterprise BDIDM adoption. However, TOE's flexibility appears to accommodate certain ambiguities. The TOE theory is arguably 'incomplete', as it does not include individual aspects of adoption, which seems critical for adopting disruptive technologies like BDIDM, especially SSI, in organisations. Therefore TOE should be extended to TOEU to include the User factors.

### 5.1. Limitations

A reflection on the research process identified some limitations linked to methodology, theory, and researchers' experience.

At the methodological level, the first limitation was the philosophical choice of positivism and quantitative methods, which did not allow for deeper insights into the current state of BDIDM adoption in the South African context. Although this was due to the relative newness of the technology, which made the study prioritise prediction over history, a different approach would yield different results. The second limitation concerned data analysis. It was impossible to test the structure model of TOE-BDIDM using path analysis, as suggested by the SEM framework. This was due to the dichotomous nature of the dependent variable as intentionally set according to the study objective.

At the theoretical level, the key limitation was that the TOE theory was found to be, to some extent, incomplete. Although it was more suitable for the context than other theories found in the literature, it did not accommodate measures of the individual aspects of BDIDM adoption in organisations, which might be essential for the smooth adoption of this disruptive technology.

At the researchers' experience level, there were some inconsistencies due to human error. The most noticeable limitation in this category was that the User Privacy item was missing from the interval-scaled data because it was unintentionally omitted in data collection. Hence, privacy was only measured on a binary scale, making it impossible to be part of the regression analysis and SEM of the measurement model.

### 5.2. Future Research

Given the limitations highlighted above, methodological, theoretical, and topical measures could be adopted for further research on the topic or in the field.

From a methodological perspective, as blockchain technology evolves, further research might consider using a different approach, including combining quantitative and qualitative data collection and analysis methods, to accommodate both depth and accuracy. Alternatively, one might want to record the actual state of BDIDM adoption in a specific context, for instance, using a case study or grounded theory research strategy rather than a survey.

From a theoretical perspective, future investigation of the adoption of disruptive technologies like BDIDM-SSI might consider combining TOE with another theoretical framework, such as TAM, to include the individual aspects of adoption in organisations. Alternatively, one might use any other theory, or develop one, that provides for all four contexts, namely: Technology, Organisation, External Environment, and 'User' (TOEU).

From a topical perspective, the reflection led to an understanding that BDIDM adoption might be beyond silos of adoption. It was learnt that, given its relatively high disrup-

tiveness, BDIDM might perhaps be impractical for sole organisations to adopt without a national strategy supporting or enforcing the adoption. For a more sustainable adoption, a government might need to be actively involved in, if not initiate, the adoption. This is especially important in the case of national BDIDM. Therefore, future research might rather study the national adoption of BDIDM, or the broader institutionalisation of blockchain.

**Author Contributions:** Conceptualization, S.M.M.; Methodology, S.M.M.; Validation, S.M.M.; Formal Analysis, S.M.M.; Investigation, S.M.M.; Resources, S.M.M.; Data Curation, S.M.M.; Writing—Original Draft Preparation, S.M.M.; Writing—Review & Editing, S.M.M.; Visualization, S.M.M.; Supervision, S.R.; Project Administration, S.M.M. All authors have read and agreed to the published version of the manuscript.

**Funding:** This research received no external funding.

**Institutional Review Board Statement:** The study was conducted in accordance with the Declaration of Helsinki, and approved by the Ethics Committee of the University of Cape Town (protocol code: REC 2020/11/003, date of approval: 13 July 2022).

**Informed Consent Statement:** Informed consent was obtained from all subjects involved in the study.

**Data Availability Statement:** The data presented in this study are accessible on figshare, at https://doi.org/10.25375/uct.20180054 (accessed on 1 August 2022).

**Conflicts of Interest:** The authors declare no conflict of interest.

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
