# Peer review of "Factors Affecting Organisations’ Adoption Behaviour toward Blockchain-Based Distributed Identity Management: The Sustainability of Self-Sovereign Identity in Organisations"

_sustainability, doi:10.3390/su141811534_

Round 1
Reviewer 1 Report
In this paper, the authors investigate how technological, organizational and environmental (TOE) factors affect BDIDM adoption in organizations using the TOE-BDIDM model. It aims to determine the most critical factors explaining the behavior while assessing TOE-BDIDM effectiveness and appropriateness. However, there are some major concerns as follows:
1. The resolution of the figures is too low (such as figure 1), which leads to the blurring of the text and affects the readability. The author should provide the clearer original figures.
2. In Table 2, What does P >. 5 mean?
3. There are too many keywords in the paper, and the author needs to further simplify.
4. In the literature, there are some works on blockchain in organizations. However, many literatures do not consider the privacy problem in the intelligent environment. Authors are suggested to review more new and relevant research to support their research contribution. Some refs could be useful, e.g., BTS: A Blockchain-based Trust System to Deter Malicious Data Reporting in Intelligent Internet of Things; A game-based deep reinforcement learning approach for energy-efficient computation in MEC systems; A privacy-protected intelligent crowdsourcing application of IoT based on the reinforcement learning.
5. There are also some typos in this paper. Please carefully go through the manuscript to improve its presentation.
6. The format of the paper shall be carefully checked.
7. Some references have incomplete compilation, e.g., missing volume/page numbering
8. Presentation is better but there is still room for improvement.
Author Response
|
Reviewer comment |
Authors’ response |
R1.1 |
The resolution of the figures is too low (such as figure 1), which leads to the blurring of the text and affects the readability. The author should provide the clearer original figures. |
We have provided files of the original figures. |
R1.2 |
In Table 2, What does P >. 5 mean? |
In Table 2, p>.05 means that Chisq should be insignificant for a better fit of the model. A significant Chisq indicates a significant difference between the hypothesized model and the actual variance matrices, thus a poorer fit. This explanation was added to the description above the table to clarify the matter. |
R1.3 |
There are too many keywords in the paper, and the author needs to further simplify. |
Redundant keywords were removed. |
R1.4 |
In the literature, there are some works on blockchain in organizations. However, many literatures do not consider the privacy problem in the intelligent environment. Authors are suggested to review more new and relevant research to support their research contribution. Some refs could be useful, e.g., BTS: A Blockchain-based Trust System to Deter Malicious Data Reporting in Intelligent Internet of Things; A game-based deep reinforcement learning approach for energy-efficient computation in MEC systems; A privacy-protected intelligent crowdsourcing application of IoT based on the reinforcement learning. |
This study focuses on blockchain-based distributed identity (BDIDM) in the context of enterprise identity management (IDM) systems. More specifically, it is about organisations’ adoption behaviour toward blockchain-based identification and authentication of users of their systems. In this regard, Self-Sovereign Identity (SII) is considered a sustainable solution from the user privacy perspective. But BDIDM could be implemented differently than in an SSI approach. Thus we also track implications for users’ privacy when the SSI is not preferred by organisations. We have added the definition of the IDM concept it is about in the first paragraph of the introduction
The study does not necessarily deal with the transparency of data reported on a given blockchain or specific data manipulation, nor is it based on intelligent systems. These might be industry-specific applications of BDIDM. Thus, only the third reference suggested by Reviewer 1 applies. We feel that the rest go off or beyond the topic. We have also added 3 recent references on SII to support our argument: Please see the 1st paragraph of section “sustainability of the SSI model in organisations”. We have also rephrased the title and some headings to reflect better what is being discussed.
Toward the end of the 2ndt paragraph of the introduction, the study does mention the link between BDIDM, IoT, and cloud computing. But it does not go into the details of these industrial applications to keep the focus on the primary concept of IDM as defined in information security terms: identification and authentication of users on a system (Whitman & Mattord, 2018). |
R1.5 |
There are also some typos in this paper. Please carefully go through the manuscript to improve its presentation. |
Additional proofreading was done. We also rearranged the content to improve the presentation |
R1.6 |
The format of the paper shall be carefully checked. |
An additional format check was done to the best of our ability |
R1.7 |
Some references have incomplete compilation, e.g., missing volume/page numbering |
We fixed in-text citation errors and some incomplete references, but we could not find some volume/page numbering (We were hoping to get the MDP editing service to assist us with this). |
R1.8 |
Presentation is better but there is still room for improvement. |
To improve the presentation, we have set some tables with “all borders”, reduced the font size in some Tables to decrease the table size, and moved some figures and tables to remove blank spaces. We also removed the Table “Skewness and peakedness of distributions” since the narrative provided reports normality testing results sufficiently. |
Reviewer 2 Report
This study looks at how technological, organizational, and environmental (TOE) factors affect BDIDM adoption in organizations that use the TOE-BDIDM model. This paper looks like a business model. However minor corrections are required:
1. In abstract, please clearly specify the Purpose, Methodology, and problem findings. Please abstract should be 1 para.
2. The overall flow is not clear highlight the major contribution in the contribution part of Introduction section.
3. Organization of the paper is written end of Introduction.
4. What is meaning Ha1, Ha2, ....., Ha10 in Fig. 1?
5. Please include 5.1. Summary of findings and 5.2. Limitations before Conclusion after the results sections.
6. It is recommended to use a professional proofread and native English correction.
Author Response
R2.1 |
In abstract, please clearly specify the Purpose, Methodology, and problem findings. Please abstract should be 1 para. |
Parts of the abstract were removed or rephrased so that the Purpose, Methodology, and problem findings are much clearer |
R2.2 |
The overall flow is not clear highlight the major contribution in the contribution part of Introduction section. |
We rephrased the contributions part in the introduction, so they reflect the studies’ major contributions more clearly. |
R2.3 |
Organization of the paper is written end of Introduction. |
We have moved the layout at the end of the Introduction as suggested |
R2.4 |
What is meaning Ha1, Ha2, ....., Ha10 in Fig. 1? |
We set “Ha” as a short form of “alternative hypotheses”. Each arrow in figure 1 represents a particular Ha, from Ha1 to Ha10. We have fixed the annotations, changing from “Ha1, Ha2” to “Ha1 & Ha2” to show that the arrow represents both Ha1 and Ha2 as discussed toward the end of section “organizational factors”. We have updated Figure 1 and the description in the first paragraph of section “The TOE-BDIDM model” accordingly. |
R2.5 |
Please include 5.1. Summary of findings and 5.2. Limitations before Conclusion after the results sections. |
This suggestion was partially addressed. We moved some of Summary of findings at the end of the results section while the rest was blended with conclusions. Our recommendations for further research are closely related to the limitations identified. Thus, we found moving Limitations far from further research to be problematic because it will disturb the flow. |
R2.6 |
It is recommended to use a professional proofread and native English correction. |
We only did our own proofreading. We did not manage to use the MDP language editing service as planned to due delays in payment from our institution’s finance service. We wish would have enough time to follow up and get this done. |
Round 2
Reviewer 1 Report
The authors have addressed my concern and it can be accepted now. I encourage the authors to upload the source code and data of the paper/experiments to a public repository once that the paper will be accepted (if applicable), and to include the link of the repository in the article.
Reviewer 2 Report
Thanks for correction.